# Non-Small-Cell Lung Cancer in 2022: A Review for General Practitioners in Oncology

**Hamid Mithoowani** [1,*] **and Michela Febbraro** [2]

1 Grand River Regional Cancer Centre, 835 King St. W., Kitchener, ON N2G1G3, Canada
2 Juravinski Cancer Center, Hamilton Health Sciences Center, 699 Concession St., Hamilton, ON L8V5C2, Canada; michela.febbraro@medportal.ca
* Correspondence: mithooh@mcmaster.ca

**Abstract:** Lung cancer is the leading cause of cancer death in Canada and a significant cause of morbidity for patients and their loved ones. There have been rapid advances in preventing, screening and treating this disease. Here, we present a contemporary review of treatment of non-small cell lung cancer in Canada based on current best practices. The focus of this review is to highlight recent data in screening for lung cancer, management of patients with early and locally-advanced non-small cell lung cancer, as well as management of patients with metastatic disease. There is a special focus on the incorporation of immunotherapy into practice and its associated toxicities.

**Keywords:** non-small cell lung cancer; lung cancer; screening; immunotherapy; iRAE; iRAEs; CAGPO; Canadian Association of General Practitioners in Oncology; ALK; EGFR; K-RAS; KRAS

## 1. Introduction

Lung cancer is the leading cause of cancer death in Canada. It can lead to significant morbidity for patients and their loved ones. There have been rapid advances in preventing, screening and treating this disease. The objective of this review is to provide an overview of non-small-cell lung cancer for the general practitioner in oncology. This review starts with the prevention and screening of lung cancer, and finishes with recent advances in targeted therapy and immunotherapy for patients with incurable disease.

## 2. Screening and Prevention

Smoking cessation is integral for the prevention of lung cancer. Public health measures aimed at quitting smoking have contributed to the reduced incidence of lung cancer [1,2]. Smoking cessation is also important in the treatment of lung cancer [3]. After establishing a lung cancer diagnosis, smoking cessation is linked to increased survival time, decreased post-operative complications, improved response to systemic therapy, improved response to radiation and improved quality of life [4]. Smoking cessation is encouraged in all active smokers and especially individuals undergoing lung cancer screening or treatment. The most effective smoking-cessation strategy for lung-cancer patients is unknown. Patients are encouraged to speak to their primary healthcare providers to discuss which strategy would best suit individual needs [5].

Clinical outcomes in lung cancer are directly related to the cancer stage at the time of diagnosis. For example, patients who present with stage I lung cancer have a five-year survival of 68.4%, whereas patients who present with stage IV lung cancer have a five-year survival of 5.8% [6,7]. Unfortunately, most lung cancers present as stage IV disease, which is associated with a poorer survival and increased symptom burden. Screening for and early detection of lung cancer significantly reduces lung-cancer mortality. The NELSON trial randomized individuals aged 50 to 74 years who actively smoke or previously smoked, in the Netherlands and Belgium, to receive low-dose CT (LDCT) or no screening at baseline, 1, 3 and 5.5 years [8]. For men, lung-cancer mortality was reduced in the screened cohort

by 24% when compared to the unscreened cohort at 10 years. For women, comprising only 16% of the study population, lung-cancer mortality was reduced in the screened cohort by 33% when compared to the unscreened cohort at 10 years [9]. This study was not powered to detect differences in all-cause mortality. Lung cancers were detected earlier in the screened cohort. A diagnosis of stage I lung cancer comprised 59% of cancers detected in the screened cohort as compared with 13.5% in the unscreened cohort [9].

There have been several other lung-cancer-screening trials completed to date. A 2020 meta-analysis of seven trials included over 84,000 patients with a minimum 15-pack-per-year smoking history. Lung-cancer mortality was reduced in patients with LDCT by 17% [10]. Lung-cancer screening has the potential to save lives and lower healthcare-system expenditures [11]. The Canadian Task Force on Preventive Health Care now recommends screening for lung cancer with three annual LDCT scans in high-risk adults aged 55 to 74 years who currently smoke or quit less than 15 years ago, with a smoking history of at least 30 packs per year [12]. Currently, there are no publicly funded lung-cancer-screening programs in Canada. However, pilot studies are ongoing to investigate the feasibility of the implementation in high-risk populations, with Ontario having the first provincial program in Canada and the province of British Columbia slated to start its provincially funded screening program in the spring of 2022 [13].

## 3. Resectable Lung Cancer

The standard of care for patients with stage I and II, as well as select patients with stage IIIA, non-small-cell lung cancer (NSCLC) is surgical resection. Following resection, patients may be offered adjuvant systemic therapy. In the Lung Adjuvant Cisplatin Evaluation (LACE) meta-analysis, patients with completely resected NSCLC received adjuvant systemic therapy with a cisplatin-based doublet regimen. The benefit of adjuvant therapy varied by stage with stage IB (tumor $\geq$ 4 cm), having a 3% decrease in the risk of death at 5 years. It should be noted that the benefit in stage IB patients was only for those with a high risk for recurrence. The benefit of adjuvant chemotherapy increased to 13% for stage III lung cancers when compared with no chemotherapy [14]. Adjuvant chemotherapy typically consists of four cycles of a cisplatin-based doublet and should be offered for completely resected stage IB (high risk) to IIIA non-small-cell lung cancers [15].

Despite the use of post-operative chemotherapy, roughly half of patients with stage IB (tumor $\geq$ 4 cm) to roughly three-quarters of patients with stage IIIA lung cancer relapse with metastatic disease [14]. Prior to 2020 there was no role for additional systemic therapy following adjuvant chemotherapy. Recently, the data regarding further adjuvant treatment with immunotherapy or oral tyrosine kinase inhibitor (TKI) therapy have challenged the current standard of care.

Osimertinib, a third-generation oral EGFR–TKI, selectively binds both EGFR driver mutations and EGFR T790M resistance mutations [16]. It is approved for the adjuvant therapy of completely resected stage II and stage III NSCLC. The phase III ADAURA trial assessed the role of adjuvant osimertinib therapy vs. placebo for up to 3 years [17]. Patients had to have completely resected stage IB (tumor > 3 cm) to IIIA NSCLC with EGFR exon 19 deletion or exon 21 L858R driver mutations. The study reported an absolute improvement in two-year disease-free survival (DFS) of 37% and a relative improvement in two-year DFS of 80% [17]. Given the efficacy of immunotherapy in stage III and IV disease, there is growing interest to apply it in the adjuvant setting. The IMpower010 demonstrated efficacy of adjuvant atezolizumab in patients with stage IB (tumors $\geq$ 4 cm) to IIIA NSCLC who had undergone surgery and up to four cycles of adjuvant chemotherapy [18]. In the primary analysis of patients with stage II–IIIA NSCLC and PD-L1 on 1% or more of tumor cells, 16 cycles of atezolizumab led to a relative improvement in 3-year DFS of 44% compared to best supportive care [18]. Several additional trials evaluating adjuvant oral TKI or immunotherapy are ongoing. The most anticipated of the Canadian studies is BR31, a phase III placebo-controlled trial studying adjuvant durvalumab in completely resected NSCLC [19].

Although data for overall survival (OS) are immature and not reported for both ADAURA and IMpower010, chemotherapy, as the sole adjuvant therapy for completely resected stage IB (tumors ≥ 4 cm) to IIIA NSCLC, may be becoming a thing of the past. The opportunity to enroll patients in clinical trials assessing the role of additional adjuvant therapy with immunotherapy or oral TKI should be the goal of all clinicians treating curative lung cancer.

## 4. Stage III Non-Small-Cell Lung Cancer (NSCLC)

Stage III NSCLC compromises approximately 20% of cases at initial diagnosis and refers to any tumors metastasized to mediastinal lymph nodes (Any T stage, N2) or large tumors that may involve local lymph nodes (T3N1 and T4N0) [20]. Stage III NSCLC is a heterogeneous disease and treatment varies depending on tumor burden, symptom burden and patient factors. Surgery for stage III NSCLC is controversial. So-called "trimodality" treatment, consisting of neoadjuvant chemotherapy and radiation, followed by surgery, can be considered in a select group of patients with single station mediastinal lymph node involvement. In the Intergroup 0139 study, patients whose disease was amenable to lobectomy demonstrated a significant survival benefit with the addition of surgery after preoperative chemotherapy and radiation. Patients requiring pneumonectomy did not demonstrate the same benefit on account of significant perioperative mortality [21]. Patient selection for surgery is crucial and should be ideally performed in a multidisciplinary format.

A majority of patients with stage III NSCLC are deemed unresectable by either patient choice, tumor burden or because they are poor surgical candidates. Multimodality treatment involving chemotherapy and radiation, either given concurrently or sequentially, has given patients the best chance at long-term survival. Survival with concurrent chemotherapy and radiation is better than if given sequentially [22]. Patients should have a good performance status and be able to tolerate multimodality therapy. Common side effects include esophagitis, hematological toxicity and pneumonitis. An alternative to patients with a borderline performance status is sequential treatment with chemotherapy, followed by radiation. With this approach, five-year survival is approximately 10% [22].

For many years, studies have tried to improve on combination chemotherapy and radiation for unresectable stage III NSCLC. Escalation of the number of chemotherapy cycles and radiation doses has failed to improve overall survival in this group of patients [23,24]. For the first time, an improvement in overall survival was demonstrated with the incorporation of immunotherapy for patients who have completed concurrent chemotherapy and radiation therapy. The most recent update of the PACIFIC trial showed that consolidation therapy with one year of durvalumab reduced the risk of death by 29% as compared to the placebo. The four-year overall survival is 49.6% for patients who received immunotherapy as opposed to 36.3% who did not [25]. A baseline CT scan should be performed post-chemotherapy/radiation to exclude radiation pneumonitis and disease progression. Treatment should be initiated within 42 days of finishing chemotherapy and radiation. There was an increased risk of thyroid dysfunction and a small but clinically relevant risk of pneumonitis with immunotherapy [26]. Durvalumab is dosed every two weeks, but administration of the agent on a 28-day cycle has been approved by certain provinces to reduce travel burden and potential exposure during the COVID-19 pandemic.

Given the poor outcomes for stage III NSCLC and high rates of local relapse, there is renewed interest in neoadjuvant strategies. As one example, patients with stage III NSCLC involving single or multiple mediastinal lymph nodes underwent neoadjuvant durvalumab immunotherapy and chemotherapy, followed by surgery. Major pathological response, defined as less than 10% of viable tumor cells at the time of surgery, was achieved in 62% of patients. An additional 10% of patients had a complete pathological response [27]. These endpoints have been correlated with overall survival, and many other studies are ongoing, looking at the incorporation of immunotherapy in the neoadjuvant setting.

If patients are not suitable for multimodality therapy, definitive radiation or palliative radiotherapy is an effective option for symptom control.

### 5. Metastatic Non-Small-Cell Lung Cancer (NSCLC)

Most patients with lung cancer present with distant metastases, and many others with early stage or locally advanced disease relapse into the metastatic setting. The goals in management of metastatic NSCLC are to improve or maintain quality of life, and to prolong overall survival. Early integration of palliative care has not only been shown to improve quality of life and reduce depression but also has been shown to improve overall survival [28]. In terms of systemic therapy, options for treatment include chemotherapy, targeted therapy and immunotherapy. All non-squamous tumors should be tested for driver mutations. These driver mutations should be suspected in individuals who do not have a smoking history or those with a light smoking history. Tumors of squamous histology diagnosed in individuals who do not have a smoking history may be considered for driver mutation on a case-by-case basis. Generally, for patients harboring a mutation in EGFR, ALK or ROS1, targeted therapy is preferred over other therapies, as it is more effective and less toxic. The International Association for the Study of Lung Cancer recommends testing for EGFR, ALK and ROS1 at a minimum, and newer guidelines also recommend testing for BRAF, KRAS, MET, NTRK and RET [29,30]. This review focuses on common driver mutations with actionable targets. For patients who do not possess a driver mutation, options for treatment include single-agent immunotherapy, combination immunotherapy regimens or chemotherapy alone. A general overview of treatment of metastatic NSCLC is found in Figure 1.

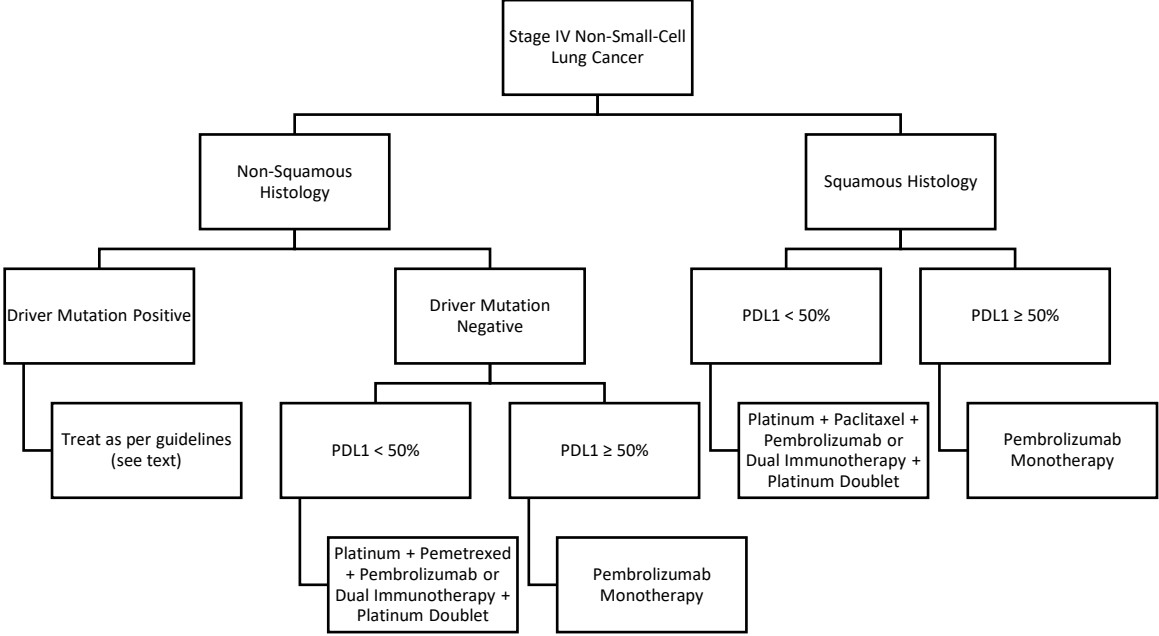

**Figure 1.** First-line treatment for metastatic non-small-cell lung cancer.

### 5.1. Chemotherapy

Chemotherapy remains a first-line option for patients who may not be eligible for single agent immunotherapy or combination immunotherapy regimens. These patients may have contraindications to immunotherapy, such as pre-existing autoimmune conditions, or there may be a concern about performance status and the risk of toxicity with combination immunotherapy regimens. Platinum doublets are typically used in this setting. For nonsquamous metastatic NSCLC, a typical example would be carboplatin or cisplatin and pemetrexed for 4–6 cycles, followed by maintenance pemetrexed until disease progression or unacceptable toxicity. For squamous metastatic NSCLC, a platinum doublet may consistent of carboplatin or cisplatin and either paclitaxel or gemcitabine.

### 5.2. EGFR Mutations

Epidermal growth factor receptor (EGFR) mutations are present in approximately 10% of all NSCLC cases in north America and Western Europe, as well as 30–50% of individuals in East Asia [31]. In advanced NSCLC, the presence of an EGFR mutation strongly predicts for response to EGFR TKIs. EGFR TKIs should be used in the first-line setting, regardless of PDL1 status, as immunotherapy has very poor response rates for EGFR mutated patients. As of 2017, osimertinib is the first line EGFR TKI treatment for patients with advanced NSCLC, with the common sensitizing EGFR mutations in exon 21 L858R and exon 19 deletions with or without concomitant T790M mutation. The FLAURA clinical trial compared osimertinib to gefitinib or erlotinib in treatment-naïve EGFR-mutated advanced NSCLC and demonstrated a 54% relative improvement in progression-free survival (PFS), as well as an absolute OS benefit of 6.8 months [16,32]. The treatment also showed greater intracranial efficacy and better tolerability with convenient once-daily oral administration. Osimertinib is associated with QTc prolongation and decreased ejection fraction, which are reversible upon discontinuation [33]. For patients with uncommon EGFR mutations, treatment with afatinib has been best studied. A post hoc analysis of LUX-LUNG 3 and LUX-LUNG 6 demonstrated a benefit with the use of afatinib for uncommon mutations [34]. The most frequently reported side effects for afatinib include diarrhea, rash, stomatitis, paronychia and dry skin.

EGFR exon 20 insertion mutations are a unique class of mutations associated with resistance to typical EGFR TKIs. Typical TKI therapy is associated with low response rates and should not be used as upfront therapy [35]. Standard first-line therapy includes platinum-based chemotherapy. Recently, phase I data have emerged for amivantamab and mobocertinib therapies after treatment with systemic chemotherapy. Amivantamab is an intravenous bispecific EGFR and MET receptor antibody and has demonstrated an overall response rate (ORR) of 40% in the phase 1 CHRYSALIS trial with median PFS of 8.3 months [36]. Mobocertinib is an oral TKI demonstrating an ORR of 28% and median PFS of 7.3 months [37].

### 5.3. ALK Rearrangements

Anaplastic lymphoma kinase (ALK) rearrangements are detected in 2–5% of all NSCLCs [38]. These rearrangements are sensitive to ALK TKIs, with prognosis measured in years, when the appropriate therapy is started immediately. Brain metastases and leptomeningeal disease are common in patients with advanced ALK positive NSCLC; therefore, first-line treatment prioritizing CNS activity is favored [39]. ALK TKIs should be used in the first-line setting, regardless of PDL1 status. First-line treatment with either alectinib, brigatinib or lorlatinib is recommended, with each of these agents demonstrating superior PFS and CNS activity when compared to earlier ALK TKIs, such as crizotinib [39].

Three studies have compared alectinib to crizotinib in the first-line metastatic setting. The Japanese study J-ALEX, the Asian study ALESIA and the global study ALEX demonstrated a 66%, 78% and 53% respective relative reduction in risk of progression or death [40,41]. OS data are still immature, with the median OS not being met for alectinib at the 5-year interim analysis of ALEX compared to 54.7 months with crizotinib [42]. The most common adverse events with alectinib include anemia, myalgia, increased bilirubin, increased weight and photosensitivity.

In the ALTA 1L study, brigatinib demonstrated improved efficacy over crizotinib in the first-line advanced setting with a 51% reduction in the risk of progression or death [43]. The overall survival data are still immature. The most common adverse events with brigatinib include elevated CK, cough and hypertension. Interstitial lung disease/pneumonitis occurred in approximately 4% of patients. Brigatinib-related pneumonitis tends to occur early in treatment necessitating close monitoring, especially within the first few weeks of starting therapy [44].

Lorlatinib is a third-generation ALK TKI and was specifically designed to cross the blood–brain barrier to achieve high CNS concentration. In the CROWN study, lorlatinib

demonstrated improved efficacy over crizotinib in the first-line setting with a 72% reduction in the risk of progression or death. Overall survival data are immature [45]. The most common adverse events with lorlatinib include hypercholesterolemia, hypertriglyceridemia, edema and peripheral neuropathy. Unique to lorlatinib is its neurologic side effects, as they relate to cognition (i.e., memory impairment) and mood (i.e., affect lability), which are reversible with dose interruptions. The decision to initiate alectinib, brigatinib or lorlatinib in Canada depends on patient characteristics and availability of the drugs. In contrast to EGFR-mutated NSCLC, immunotherapy is typically associated with poor response rates in this patient population.

*5.4. ROS1 Rearrangements*

ROS1 is a receptor tyrosine kinase of the insulin receptor family and is present in approximately 1–2% of patients with NSCLC [46]. These rearrangements are sensitive to the ROS1/MET inhibitor crizotinib. Evidence supporting the use of crizotinib for ROS1 positive NSCLC is based on early phase clinical trials demonstrating an ORR of 72%, with median duration of response of 25 months [47]. Lorlatinib, which is approved in ALK-positive NSCLC, appears to overcome acquired resistance to crizotinib in patients with ROS1 positive NSCLC [48]. Entrectinib, an oral ROS1/TRK inhibitor, has also been approved in Canada for the treatment of ROS1 positive NSCLC.

*5.5. KRAS Mutations*

Kirsten rat sarcoma viral oncogene homolog (KRAS) mutations have been identified in several cancers. It is the most common driver mutation in lung cancer. Mutations in codon 12 (KRAS G12C mutations) comprise almost 50% of KRAS mutations and, in contrast to other mutations, they often occur in individuals with a smoking history [49]. Once thought to be an undruggable target, emerging KRAS inhibitors are showing promise in the treatment of metastatic NSCLC [50]. Sotorasib is the first targeted agent developed for KRAS G12C mutations. It specifically and irreversibly inhibits KRAS G12C by locking it in an inactive GDP-bound state [51]. In a phase I study of previously treated NSCLC patients with KRAS G12C mutations, an ORR of 32% and a disease control rate of 88% were seen with sotorasib [50]. The most common adverse events were liver-function-test abnormalities, diarrhea, anemia, hepatitis and hyponatremia. The CodeBreak 200 trial will be evaluating sotorasib versus docetaxel in NSCLC patients who have progressed on at least one prior systemic therapy, with past treatments having to include platinum-based chemotherapy and an immune checkpoint inhibitor [52].

*5.6. Immunotherapy*

Immunotherapy has significantly changed the landscape in management of patients with metastatic NSCLC. In 2015, the first large study of immunotherapy demonstrating activity in NSCLC was reported in the Phase II Checkmate 063 study. In heavily pre-treated patients, nivolumab demonstrated meaningful benefit and acceptable toxicity [53]. In the short years that have followed, multiple immunotherapy strategies have been developed. Long-term survival can be achieved for some patients with metastatic NSCLC, a so-called "tail of the survival curve" [54].

Responses to immunotherapy depend on tumor PDL-1 expression. PDL-1 expression is broadly categorized in three groups: PDL-1 negative (<1% of tumor cells express PDL-1), PDL-1 low positive (1–49%) and PDL-1 positive (>50%). Survival and duration of response increase as PDL-1 expression increases. For patients with PDL-1 positive tumors, single-agent immunotherapy has been consistently shown to be superior to chemotherapy, both in toxicity and in improving survival. Median survival with pembrolizumab in the KEYNOTE-024 study reached 26.3 months, and, notably, 31.9% of patients reached the five-year survival mark, the highest to date in a phase III study [55]. Similar studies for PDL-1 positive patients have shown immunotherapy agents atezolizumab and cemiplimab

to be more effective than chemotherapy [56,57]. These studies demonstrated that, for the first time, chemotherapy can be avoided for some patients with metastatic NSCLC.

For patients that have PDL-1 negative (<1%) or PDL-1 (1–49%) tumors, newer combination strategies have become standard of care in the first line setting. The KEYNOTE-189 study randomized patients with metastatic nonsquamous NSCLC to carboplatin and pemetrexed or the same regimen plus pembrolizumab. The combination of chemotherapy and immunotherapy yielded an overall survival of 22 months and reduced the risk of death by 44% compared to chemotherapy alone [58]. Similarly, the combination of chemotherapy and pembrolizumab reduced the risk of death by 36% in patients with metastatic squamous NSCLC in the KEYNOTE-407 study. In this study, the chemotherapy consisted of carboplatin and a taxane [59]. Dual immunotherapy combination trials have also been reported for metastatic NSCLC. In the Checkmate-9LA study, patients with nonsquamous or squamous histology were randomized to receiving a platinum doublet or ipilimumab and nivolumab, plus two cycles of a platinum doublet. The dual immunotherapy–chemotherapy arm yielded a median overall survival of 15.8 months and reduced the risk of death by 28% compared to chemotherapy alone [60]. Dual immunotherapy combinations have not been compared head-to-head with chemotherapy–immunotherapy combinations, and it remains to be seen with longer follow-up whether certain subgroups will benefit from one regimen over another.

### 5.7. Recognition and Management of Immune Related Adverse Events (iRAEs)

With immunotherapy being incorporated in all stages of lung-cancer treatment, recognition and management of immunotherapy toxicity deserves special attention. Chemotherapy toxicity is different from iRAEs in three important ways.

Firstly, the types of toxicities seen with immunotherapy are different. The most common iRAEs with single-agent immunotherapy are rash, pruritus, diarrhea and thyroid dysfunction [61]. Immunotherapy is usually continued for mild rashes that do not impair quality of life and are treated with moderate strength topical corticosteroids (e.g., hydrocortisone 1–2.5% applied b.i.d). Moderately severe rashes that are symptomatic or cover a larger surface should be treated with a two-week course of tapering steroids (e.g., prednisone 0.5 to 1 mg/kg). A severe rash causing sloughing should prompt IV steroids (e.g., methylprednisone 1 mg/kg) and admission to hospital for supportive management [62]. Thyroid dysfunction is commonly seen with immunotherapy agents and treatment can generally be continued with adequate thyroid replacement. Often, a period of asymptomatic thyroiditis with a depression in TSH levels is observed before a period of hypothyroidism. Hyperthyroidism with immunotherapy is much less common. Thyroid function should be checked every 3–6 weeks on treatment. If both the TSH and T4 levels are low, a central cause of hypothyroidism, such as hypophysitis, should be considered.

Serious iRAEs deserve mention, as they can cause mortality. Fortunately, events causing mortality are rare and range from 0.36% for single agent immunotherapy to 1.23% for combination immunotherapy treatments. The two most common iRAEs causing mortality are colitis and pneumonitis. [63]. Early recognition and prompt administration of steroids can decrease risk of fatal events. Colitis should be suspected in patients with an increase of at least four bowel movements over baseline, sometimes associated with cramping, bloating or hematochezia. For moderate symptoms, oral corticosteroids should be prescribed, and if patients are not responding to therapy within 48 h, additional immunosuppression with a TNF-alpha inhibitor may be indicated [64]. Establishing a diagnosis of pneumonitis is challenging, as the symptoms are relatively non-specific and can be mimicked by disease progression, pneumonia, pulmonary embolism or radiation pneumonitis. A dry cough and progressive dyspnea are the most common presenting symptoms [65]. The combination of symptoms and ground glass opacities occurring outside radiation fields or other diffuse changes on CT scan generally suffice to establish a diagnosis. Some experts advocate for bronchoscopy to rule out other causes, but the diagnosis is clinical, and there is no definitive histopathology [66]. Management is similar to other iRAEs. Moderate symptoms can be

treated with oral steroids, and severe symptoms that require hospitalization are treated with IV steroids. A slow taper of at least 6 weeks is recommended, as well as follow-up imaging around the same time, to ensure resolution or stabilization of findings [65,66]. There are multiple guidelines for treatment of iRAEs, including Canadian guidelines, such as one published by Cancer Care Ontario that is referenced below [67]. ASCO and ESMO have also published clinical practice guidelines outlining specific management strategies for immune-related adverse events [62,66].

Secondly, iRAEs are less predictable than chemotherapy. Immunotherapy can essentially affect any organ. It is important to be vigilant with any change in symptoms or biochemical changes in patients undergoing immunotherapy, as toxicities can occur at any point. Most iRAEs toxicities happen in the first three months of initiation of therapy, but later toxicities can occur, and first-onset iRAEs are even possible well after stopping immunotherapy. Toxicities that usually present early in the course of treatment include rash, pruritis and diarrhea/colitis. Later toxicities include hypophysitis and hepatitis [62]. The addition of a CTLA4 inhibitor, such as ipilimumab, increases the risk of severe iRAEs, as well as the duration of toxicity, and may require longer tapering of steroids [66]. CTLA4 inhibitors, in particular, increase the risk of colitis and hypophysitis [68].

Finally, treatment of immunotherapy toxicities is an active process. To manage the side effects of chemotherapy, oncologists often use passive forms of management, such as dose reductions, treatment breaks and supportive management. Immunotherapy agents have long half-lives, and typically dose reductions and treatment breaks are not routinely prescribed for the sole purpose of managing side effects. For patients with serious iRAEs, prompt immunosuppression is crucial to reduce morbidity and, in some cases, mortality. As experience grows with immunotherapy, there is emerging data that some patients who develop iRAEs may have better survival outcomes than those who do not. Although the data are mainly retrospective, there may be a group of patients who develop severe or multiple iRAEs who do not need to be retreated and have very good long-term outcomes [69,70].

## 6. Conclusions

The landscape of treating lung cancer in Canada has undergone rapid transition in the past decade. There has been success in identifying new targets that have provided meaningful benefit to patients with metastatic NSCLC, particularly in individuals who do not have a smoking history. The incorporation of immunotherapy has changed the standard of care for patients living with metastatic NSCLC, and is now being incorporated into earlier stages. Physicians treating lung cancer must now recognize and manage the unique set of toxicities associated with immunotherapy. This review only captured some of the complexities in treating NSCLC and did not, for example, go into detail about sequencing therapies. Despite these advances, lung cancer remains a significant burden of morbidity and mortality on the Canadian population. Smoking cessation and screening high-risk individuals are essential tools to ease that burden.

**Author Contributions:** Conceptualization, H.M. and M.F.; writing—original draft preparation, H.M. and M.F.; writing—review and editing, H.M. and M.F. All authors have read and agreed to the published version of the manuscript.

**Funding:** The research received no external funding.

**Conflicts of Interest:** We have read and understood Current Oncology's policy on disclosing conflicts of interest, and we declare the following interests: H.M. has participated in advisory board meetings for AstraZeneca, Pfizer, Bristol Myers Squibb and Takeda.

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
