# Peer review of "Non-Small-Cell Lung Cancer in 2022: A Review for General Practitioners in Oncology"

_curroncol, doi:10.3390/curroncol29030150_

Round 1

Reviewer 1 Report

Great review article, highlighting the evolution of the treatment of NSCLC. Here are my suggestions:

1) "Smokers" - please use a different term or phrase as smokers is associated with stigma and blame. An example would be people who smoke or use tobacco

2) lines 34-35 - "no good quality evidence on most effective cessation strategy". This is somewhat misleading for me as I am interpreting there is no way to help people with smoking cessation. Perhaps add a sentence or two about tools to help people

3) line 60 - no publicly funded screening program but should mention B.C. is slated to start their provincially funded screening program in Spring 2022

4) LACE meta-analysis. I think there should be a mention of the caveat of using adjuvant chemo in stage IB disease. The HR was 0.78-1.10, so perhaps clarify IB only if considered high risk for recurrence

5) ADAURA - you summarized the study well, but clarify it is only approved by pCODr for stages II to III, not IB

6) line 149 - durva can be given "4-weekly", not monthly

7) line 172 - "all [non-squamous] tumours should have biomarker testing. Those with squamous histology but are never smokers may consider having biomarker testing as there are rare cases of EGFR mutation found in these patients. Of course this is assessed case by case and policies may differ among provinces

8) line 218 - typo => "sued" should be "used"

9) line 118 - typo => select group [of] patients

10) brigatinib - last comment about 4% risk for pneumonitis and interstitial lung disease, perhaps add a comment about the likelihood for these AEs to occur early on with treatment initiation and patients should be monitored closely

11) you mentioned IASLC guidelines recommend a minimum biomarker test for EGFR, ALK and ROS1 but you did not discuss treatment options for ROS1. It is rare, but there are treatment options that perhaps can be briefly mentioned.

Author Response

Reviewer 1 comments:

  • We no longer use the term ‘smokers’ and instead using terms such as ‘individuals who do not have a smoking history’ or ‘individuals who currently smoke or have a previous history of smoking’
  • Line 34-35 has been revised to explain that patients should consult their primary healthcare providers regarding which smoking cessation strategy is most suitable for their personal needs
  • British Columbia’s lung cancer screening program start date has been mentioned for the readers
  • Regarding the LACE meta-analysis – further information regarding stage IB high risk patients was noted
  • Osimertinib approval for stage II and III completely resected NSCLC was added
  • The notation of durvalumab being administered on a ‘monthly’ basis was changed to every 28-day administration
  • As per reviewer suggestion, it was mentioned that all non-squamous tumors should receive biomarker testing with consideration of biomarker testing in squamous histology especially in individuals who do not have a smoking history
  • Information regarding early onset of pneumonitis with brigatinib was added
  • Typos on lines 118 (select group [of] patients), 218 (sued) were corrected – thank you!

Reviewer 2 Report

The authors present a literature review of recent data regarding current practices and management of non-small cell lung cancer (NSCLC). Although the article is well written and clear, I feel that the manuscript in its current form is not suitable for publication at this time. I think that the scope of this article, i.e. the treatment of NSCLC in its entirety, is too broad for it to be summarized in a concise literature review and be of educational value. The authors are advised to select one or a few subtopic(s) regarding treatment modalities of NSCLC and to expand on these topics in more detail. 

Author Response

The authors thank you for your feedback regarding this review. It is challenging to define the scope of a review for lung cancer when we consider the audience of general practitioners in oncology, and the authors certainly agree that it is a broad topic. In choosing the scope, we spoke to general practitioners in oncology informally. Some managed their own practices and were site-specific (i.e, treated lung or breast cancer specifically). Others practiced along side a medical oncologist in an academic or community centre, and others co-managed patients with a medical oncologist at another site, for example, in a rural community where care could be delivered closer to home. Keeping in mind the different roles that a GPO may have, we decided upon a broad review of common practices in lung cancer with the hopes that anyone looking for information could go to the references for further details. References were up to date with regards to the most recent data from clinical studies. We also highlighted some emerging data (ie, adjuvant treatment with immunotherapy, adjuvant TKI therapy, KRAS mutation studies, neoadjuvant data for stage III lung cancer) to hopefully give a sense of where the field of thoracic oncology was headed. We also included some practical tips and references for managing side effects for immunotherapy to inspire confidence in managing toxicities for general practitioners who may not be using immunotherapy on a regular basis. I'm hopeful that a general practitioner in oncology will be able to gain something from this article and enjoy the references we provided. 

Reviewer 3 Report

I think this is a very good review. I have a lot of comments, but they are all related to details. I would like to congratulate the authors on excellent work.

Comments:

1) Abstract and line 20 - Change "among the deadliest". By mortality, lung cancer causes more deaths than the next three cancer combined (Canadian Cancer Statistics 2021). If the authors mean, among the shortest median survival, then state that.

2) line 22 should read "screening for"

3) line 38-39 - These are optimistic numbers. The 2020 Canadian report on lung cancer shows a 3-year net survival of 71% for stage I and 49% for stage II. (Canadian Cancer Statistics 2020 Special Report). SEER data (USA) show 5-year period survival in stage I is 68.4% and stage II 45.1%. (JAMA Oncology 2021)

4) line 60 - BC has announced a program. http://www.bccancer.bc.ca/screening/lung

5) line 79-80 - Should be "relapse with metastatic disease"

6) line 118 - "select group of patients"

7) line 130 - Modern estimates from clinical trials are approximately 33% without durvalumab and 43% with durvalumab. (PACIFIC 5y update). 5y OS 32.1% in RTOG 0617. On a population basis, a Canadian study suggested 5y OS of 19% (Am J Clin Oncol 2020;43:615–620). 15% is not wrong, but more context might be useful since trial eligible patients are doing much better.

8) line 131-132 - add pneumonitis

9) line 141 - "and immunotherapy". Delete or rephrase - makes it sound like IO was given during the cCRT.

10) line 154 - durvalumab is misspelled

11) line 170 - "improve quality of life and reduce depression"

12) line 172 and 179 - replace "biomarker" with "driver mutation"

13) line 218 - "sued" should be "used"

14) line 266 - should be "common driver mutation"

15) line 271 - "irreversible" should be "irreversibly"

16) line 291 - "broadly categorized in the three broad groups" - eliminate either broadly or broad

17) line 300-301 - Could also mention that adding ipilimumab to pembrolizumab was not helpful. KEYNOTE-598.

18) line 314 - instead of saying "ipilimumab-nivolumab arm", I would call it the dual immunotherapy-chemotherapy arm.

19) line 325 - I would add a sentence referencing immune AE management guidelines from ASCO and ESMO.

20) Figure 1, squamous, PDL1<50% - instead of "Dual immunotherapy", should be "Dual immunotherapy + platinum doublet" or "Dual IO + Chemo"

Author Response

Reviewer 2 comments:

  • Abstract changed as per reviewer suggestions to state that lung cancer is the leading cause of cancer death in Canada
  • 5-year survival was updated as per the JAMA Oncology publication referenced
  • BC screening program was added in the screening section as per reviewer 1 comments as well
  • -Line 130: Generalized the statements regarding concurrent chemo-radiotherapy without specific information about survival. Agree – modern trials have demonstrated longer survival than the Auperin meta-analysis
  • Pneumonitis added as a common toxicity for concurrent chemo-radiotherapy
  • Regarding PACIFIC – there was an error whereby the sentence made it seem as though chemotherapy was given with immunotherapy. This line was amended to state concurrent chemotherapy and radiation therapy.
  • Grammar/wording changes to line 22 (screening for), line 78-80 (relapse to metastatic disease), line 170 (improve quality of life and reduce depression), line 172/179 (biomarker was replaced with driver mutation), line 266 (most common [driver] mutation), line 291 (broad was eliminated) were made – thank you!
  • Typos on lines 154 (durvalaumab), line 271 (irreversible) was corrected – thank you!
  • Figure 1 amended as per reviewer suggestion

Round 2

Reviewer 1 Report

Thank you for addressing the suggested changes. 

Author Response

Thank you!

Reviewer 3 Report

One remaining issue.

In the new sentence - "Tumors of squamous histology diagnosed in individuals who do not have a smoking history may be considered for driver mutation on a case-by-case basis"

Should read "considered for driver mutation testing on a case-by-case basis"

Author Response

Thank you! This sentence has been revised and uploaded
